# A novel prevascularized tissue-engineered chamber as a site for allogeneic and xenogeneic islet transplantation to establish a bioartificial pancreas

**Yanzhuo Liu**[1☯], **Maozhu Yang**[2☯], **Yuanyuan Cui**[1], **Yuanyuan Yao**[1], **Minxue Liao**[1], **Hao Yuan**[1], **Guojin Gong**[3], **Shaoping Deng**[2], **Gaoping Zhao**[1,2]*

1 Department of Gastrointestinal, Sichuan Academy of Medical Sciences & Sichuan Provincial People's Hospital, School of Medicine, University of Electronic Science and Technology of China, Chengdu, Sichuan Province, China, 2 Organ Transplantation Translational Medicine Key Laboratory of Sichuan Province, Sichuan Academy of Medical Sciences & Sichuan Provincial People's Hospital, Chengdu, Sichuan Province, China, 3 Department of Gastrointestinal Surgery, Xi Chang People's Hospital, Xi Chang, Sichuan Province, China

☯ These authors contributed equally to this work.
* gzhao@uestc.edu.cn

**Data Availability Statement:** All relevant data are within the manuscript and its Supporting information files.

## Abstract

Although sites for clinical or experimental islet transplantation are well established, pancreatic islet survival and function in these locations remain unsatisfactory. A possible factor that might account for this outcome is local hypoxia caused by the limited blood supply. Here, we modified a prevascularized tissue-engineered chamber (TEC) that facilitated the viability and function of the seeded islets in vivo by providing a microvascular network prior to transplantation. TECs were created, filled with Growth Factor-Matrigel™ (Matrigel™) and then implanted into the groins of mice with streptozotocin-induced diabetes. The degree of microvascularization in each TECs was analyzed by histology, real-time PCR, and Western blotting. Three hundred syngeneic islets were seeded into each chamber on days 0, 14, and 28 post-chamber implantation, and 300, 200, or 100 syngeneic islets were seeded into additional chambers on day 28 post-implantation, respectively. Furthermore, allogeneic or xenogeneic islet transplantation is a potential solution for organ shortage. The feasibility of TECs as transplantation sites for islet allografts or xenografts and treatment with anti-CD45RB and/or anti-CD40L (MR-1) was therefore explored. A highly developed microvascularized network was established in each TEC on day 28 post-implantation. Normalization of blood glucose levels in diabetic mice was negatively correlated with the duration of prevascularization and the number of seeded syngeneic islets. Combined treatment with anti-CD45RB and MR-1 resulted in long-term survival of the grafts following allotransplantation (5/5, 100%) and xenotransplantation (16/20, 80%). Flow cytometry demonstrated that the frequency of CD4+Foxp3-Treg and CD4+IL-4+-Th2 cells increased significantly after tolerogenic xenograft transplantation, while the number of CD4+IFN-γ-Th1 cells decreased. These findings demonstrate that highly developed

**Funding:** This study was supported by grants from the National Natural Science Foundation of China (nos. 81172832 and 81771723), Sichuan Youth Science and Technology Foundation (no. 2013JQ0020), and Special Program for Sichuan Youth Science and Technology Innovation (no. 2014TD0010).

**Competing interests:** The authors have declared that no competing interests exist.

microvascularized constructs can facilitate the survival of transplanted islets in a TECs, implying its potential application as artificial pancreas in the future.

## Introduction

Cellular transplantation represents an attractive treatment strategy for a variety of diseases, including diabetes, myocardial ischemia, and metabolic liver disease [1]. It has been reported that intrahepatic transplantation of donor-derived pancreatic islets is a realistic alternative cellular therapy for insulin-dependent diabetes mellitus [2]. Nevertheless, the procedure remains unsatisfactory due to inadequate glucose control. Although it has been proposed that the 'Edmonton Protocol' should constitute the standard guidelines for islet transplantation, having achieved high success rates [3], transplantation of islets via infusion through the portal vein results in massive islet loss within the first 2–4 d post-transplantation, and additional islets from 2–3 donors are required to achieve normoglycemia. The principal reason for this loss is islet apoptosis that occurs during the process of engraftment, which occurs following an instant blood-mediated inflammatory reaction (IBMIR) against the graft in the context of a lack of vascularity to supply oxygen [4, 5].

Anatomically, the pancreatic islets are immensely vascularized, allowing secretion of insulin and a rapid response to hyperglycemia [6]. A number of research studies have demonstrated that the islets lose their natural vascularization and extracellular matrix when isolated from the pancreas using traditional methods. These specialized characteristics render the islets highly susceptible to apoptosis due to insufficient partial pressure of oxygen and diffusion of nutrients [7, 8]. Therefore, the key to improving islet survival is to ensure rapid vascularization following transplantation and integration into the host's systemic vasculature. In addition, it has been documented that the process of revascularization begins after 2–4 d, followed by creation of a network of vessels 10–14 d after transplantation [9]. However, the density of new blood vessels after transplantation is considerably lower than that observed in natural islets, regardless of whether the islets are delivered to the liver or to extrahepatic sites [10, 11].

Subcutaneous sites have been suggested as alternatives for islet transplantation, as the grafts can be easily monitored by imaging [12] and removed for retransplantation [13, 14]. However, unmodified subcutaneous sites for islet engraftment have not demonstrated a reversal of hyperglycemia in mice or humans [15]. Inadequate vascularization not only fails to supply sufficient oxygen tension but also impedes the secretion of insulin from the islets to accomplish glucose homeostasis [16, 17]. Therefore, the construction of a prevascularized network in advance is critical for successful subcutaneous islet transplantation. Additionally, islet transplantation into subcutaneous tissues has been dependent on methods using modified biomaterials, including oxygen generators, polymers, meshes, encapsulation devices, matrices, and growth factors [18, 19]. However, such strategies almost always fail due to the development of an inflammatory response caused by severe tissue incompatibility [20]. Hence, a successful subcutaneous graft should (i) have sufficient tissue capacity; (ii) facilitate minimally invasive methods of transplantation; (iii) establish a vascular network to ensure adequate nutrition for the graft following transplantation; (iv) allow dynamic connectivity between the graft and systemic circulation; and (v) elicit minimal inflammation to reduce the host response and promote long-term graft survival [21, 22].

The objective of this study was to demonstrate that a TEC filled with Growth Factor-Matrigel™ (Matrigel™) embedded into the groin of a mouse with streptozotocin-induced diabetes

was capable of inducing local neovascularization to support the function of the allogeneic or xenogeneic islets transplanted inside. This is central to the advancement of clinical transplantation protocols because of the absence of donor organs. In addition, flow cytometry was used to characterize the inflammatory response, which is vital for determining whether a TEC can facilitate the long-term survival of islets following transplantation.

## Materials and methods

### Animals

Male BALB/c or C57BL/6J mice and Sprague-Dawley (SD) rats were used as pancreatic donors, and male C57BL/6J mice were used as recipients. All animals were 6–8 weeks old. Their cages were housed under a 12-h day/night cycle with *ad libitum* access to food and water. All procedures were compliant with the animal care principles approved by the Research Animal Protection Committee of Sichuan Provincial People's Hospital.

### Murine subcutaneous tissue engineering model

The tissue engineering model was established using a method similar to that described previously [23, 24]. Briefly, mice were anesthetized intraperitoneally using pentobarbital sodium (100 mg/kg) and placed in a supine position on gauze. The abdomen was shaved and wiped with an iodophor to sterilize the skin. A 7 mm incision was created in the groin, after which the site was prepared for insertion of a 5 mm-long silicone cylinder chamber (Dow-Corning, Harrodsburg, KY, USA) with an inner diameter of 3.35 mm (approximate volume: 44 μL). The silicone cylinder chamber was cut longitudinally and implanted adjacent to the superficial epigastric vessel in the groin, incorporating the inguinal fat pad into the distal end of the chamber. Matrigel™ (BD, Franklin Lakes, NJ, USA) and heparin (80 U/mL) (Pharmacia & Upjohn, Somerset, NJ, USA) were injected into the chamber, after which it was completely sealed with bone wax.

### Transplantation

Diabetes was induced in C57BL/6 mice through intraperitoneal injection of streptozotocin (200 mg/kg, Sigma Aldrich, St. Louis, MO, USA), which was confirmed by blood glucose levels > 400 mg/dL (18.8 mmol/L) for at least 2 consecutive days using blood glucose monitoring (Roche, Mannheim, Germany). The mice were euthanized by $CO_2$ asphyxiation. Islets (from C57BL/6 or Balb/c mice or SD rats) were isolated via cold digestion with 1.5 mg/ml collagenase (Roche, Mannheim, Germany) and then purified via discontinuous Ficoll gradients (densities: 1.11, 1.096, 1.066) of the pancreatic digests. The islets were transplanted into TECs as described previously [23, 24]. A functioning graft was defined as one in which a nonfasting blood glucose level <200 mg/dL was obtained, and rejection was defined as occurring when blood glucose levels of >200 mg/dL were observed on at least 2 consecutive days. The mice in all groups were monitored daily for the first two weeks and then once per week following transplantation until the mice were sacrificed. The TECs in diabetic mice that had remained normoglycemic for the whole 90 d were removed for evaluation of islet function.

### Intraperitoneal glucose tolerance test (IPGTT)

Recipient mice were subjected to an intraperitoneal glucose tolerance test (IPGTT) at 90 d post-transplantation to further assess metabolic capacity. The mice were fasted for 12 hours before receiving intraperitoneal D-glucose (2 mg/g in saline). Blood samples were obtained from the tail vein of the recipient mice 0, 15, 30, 60, 90 and 120 minutes after injection. Blood glucose levels were analyzed, and the different transplant groups were compared.

## Histology

Intact tissue from the TEC was removed and fixed overnight in 4% formaldehyde prior to processing and embedding in paraffin. Five-micrometer-thick sections were cut and stained with a primary antibody against either vascular endothelial growth factor-positive (VEGF) (1:250; Abcam, Cambridge, MA, UK) or CD31 (1:250; Abcam, Cambridge, MA, UK) and counterstained with hematoxylin and eosin (H&E). After washing, the tissue sections were incubated with a secondary antibody, 488-Alexa anti-rabbit or anti-guinea pig IgG (Molecular Probes, Eugene, OR, USA), at room temperature. Photomicrographs were acquired using a digital camera (BA200, Xiamen, China) with an appropriate filter.

## Quantitative real-time PCR

Relative mRNA expression levels of VEGF were quantified by qRT-PCR. Total RNA was extracted from the adipose tissue in the TEC with TRIzol reagent (Invitrogen, Carlsbad, CA, USA). First-strand cDNA was synthesized using a high-capacity cDNA reverse transcription kit (Roche, Mannheim, Germany). Subsequently, the relative expression levels of mRNA transcripts of interest were quantified using a SYBR Green kit and normalized to the expression of β-actin. The change in relative expression compared with the control (mRNA levels defined as a value of 1) was calculated using the $2^{-\triangle\triangle Ct}$ method. Primers for qRT-PCR were designed and synthesized by Qingke Biotech (Wuhan, China), and the sequences were as follows:

β-actin: forward 5′–CATCCGTAAAGACCTCTATGCCAAC–3′,

reverse 5′–ATGGAGCCACCGATCCACA–3′;

VEGF: forward 5′–AAGAGAAGGAAGAGGAGAGG–3′,

reverse 5′–GGTAGACATCCATGAACTTGA–3′

## Western blotting (WB)

TEC adipose tissue was homogenized and lysed using high-efficiency RIPA buffer (Solarbio, Beijing, China) or an NE-PER nuclear and cytoplasmic extraction kit (Boster, Pleasanton, CA, USA) with protease and phosphatase inhibitors (Roche, Mannheim, Germany). Equal quantities of proteins were separated using sodium dodecyl sulfate-polyacrylamide gel electrophoresis (10%) followed by transfer to PVDF membranes (Millipore, Burlington, MA, USA). The blots were blocked with 5% nonfat milk (BD, Franklin Lakes, NJ, USA) or 5% BSA (Amresco, Houston, TE, USA) dissolved in 1× TBST (blocking buffer). The blots were then probed with a primary antibody against VEGF (1:2000; Abcam, Cambridge, MA, UK) dissolved in blocking buffer, followed by alkaline phosphatase-conjugated mouse IgG (1:5000; Abcam, Cambridge, MA, UK) as the secondary antibody. HRP (Millipore, Burlington, Massachusetts, USA) chemiluminescence signals were detected using a chemiluminescence imaging analysis system (Tanon, Shanghai, China). Densitometry of the immunoblot images was performed using ImageJ software (National Institutes of Health, Bethesda, MD, USA).

## Immunotherapy

Mice were treated by intraperitoneal injection of 100 μg of an anti-mouse CD45RB antibody (Bio X Cell, West Lebanon, NH, USA) on days 0, 1, 3, 5, and 7 after transplantation. In addition, 500 μg of an anti-mouse CD40L antibody (MR-1, Bio X Cell, West Lebanon, NH, USA) was administered intraperitoneally on days 2, 4, 7, and 14 following transplantation.

## Cell stimulation and flow cytometry

The method of mouse euthanasia was described previously. Single cells from the spleen were suspended in RPMI 1640 culture medium (HyClone, Logan, UT, USA) supplemented with 10% fetal bovine serum (Gibco, Gaithersburg, MD, USA) and Cell Stimulation Cocktail (eBioscience, San Diego, CA, USA) and then cultured in 6-well plates for 5 hours in a 37˚C/5% $CO_2$ incubator.

Lymphocytes were separated from the spleen by passing through a 70 μm nylon mesh. Erythrocytes were lysed with ammonium chloride buffer, after which the remaining cells were washed and counted using a hemocytometer. One million cells were suspended in staining buffer with the following fluorochrome-conjugated antibodies: CD4-FITC (mAb; BD, Franklin Lakes, NJ, USA), CD25-APC (mAb; BD, Franklin Lakes, NJ, USA), IL-4-PECy7 (mAb; BD, Franklin Lakes, NJ, USA), IFN-γ-PE (mAb; BD, Franklin Lakes, NJ, USA), and Foxp3-PE (mAb; BD, Franklin Lakes, NJ, USA). Intracellular Foxp3, IL-4, and IFN-γ within lymphocytes were stained using a fixation/permeabilization staining kit (eBioscience, San Diego, CA, USA). All samples were analyzed using a FACSCanto II flow cytometer (BD, Franklin Lakes, NJ, USA) and analyzed using Flow Jo analysis software (Tree Star Inc.).

## Statistical analysis

Data were analyzed using GraphPad Prism software version 5 (GraphPad version 5.01). Graft survival in the experimental groups was compared using Kaplan–Meier survival curves. Other differences between experimental groups were evaluated using Student's t-test. $P < 0.05$ was considered statistically significant.

## Results

### Significant prevascularization of the TECs suggests that it is a suitable site for islet transplantation

The procedure for construction of the TEC is presented in Fig 1A and 1B. Following its implantation in the groin, it exhibited good histocompatibility and safety during the study. None of the mice suffered leg paralysis, wound infection, chamber exposure, or sinus formation. The chamber grafts in each TEC (>28 d) were richly vascularized, as can be observed in the macroscopic images (Fig 1C). Based on our knowledge, we hypothesized that the scale of neovascularization in the TECs was related to the duration of prevascularization. Therefore, an extensive analysis of the chamber grafts from each TEC was undertaken after 7, 14, 28, and 35 d. As shown in Fig 2A, HE-stained adipose tissue from the TEC demonstrated significantly greater formation of vessels on days 28 and 35 than on 7 d and 14 d implantation. As expected, the TEC implants stained as VEGF+ and platelet endothelial cell adhesion molecule-1-positive (CD31+) on days 7, 14, 28 and 35 post-implantation. VEGF and CD31 are used as markers of neovascularization to represent the prevascularization of the TECs. The associated vasculature of the TECs had increased significantly by days 28 and 35 (Fig 2A).

Furthermore, the qRT-PCR results demonstrated that VEGF mRNA expression levels were significantly higher on days 28 and 35 than on days 7 and 14. However, there was no difference in expression levels between 28 and 35 d (Fig 2B). Quantification of VEGF protein expression revealed trends consistent with the results of qRT-PCR (Fig 2C), and the extent of prevascularization closely correlated with the duration.

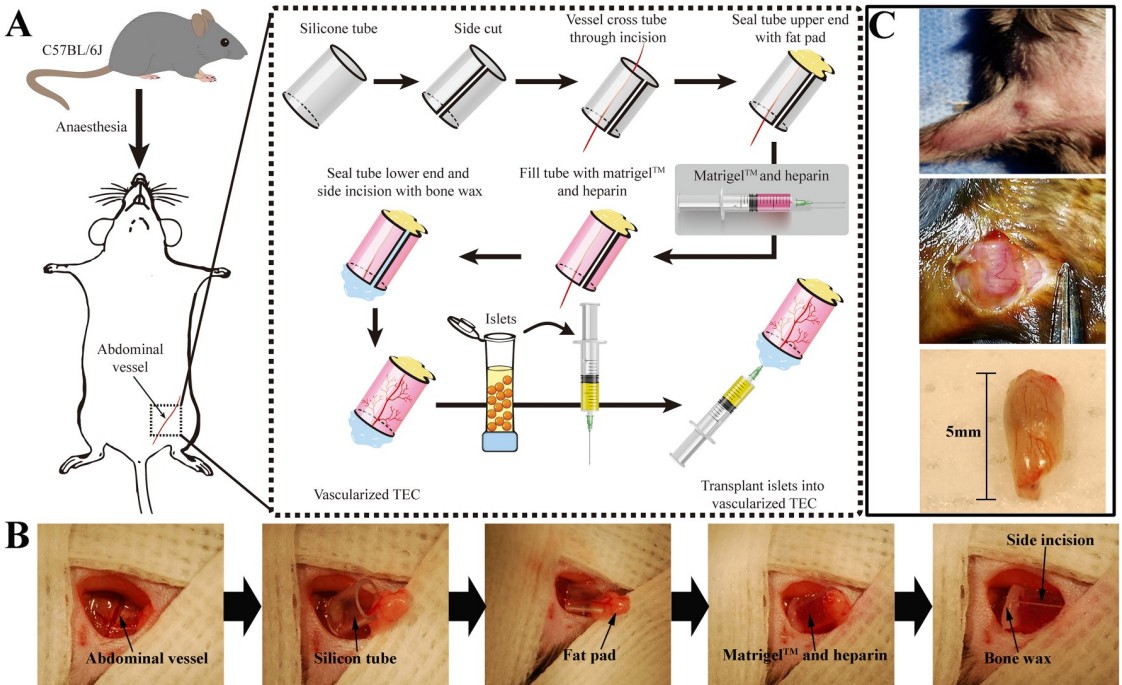

**Fig 1. Construction of tissue-engineered chamber and islet transplantation.** (A) Flow diagram of TEC construction and islet transplantation. (B) The silicone cylinder chamber, including Matrigel™ and heparin, was inserted and implanted adjacent to the superficial epigastric vessel. A fat pad and bone wax were used to close the ends of the silicone tube. (C) A microvascular network was established (50×).

## Euglycemia levels in diabetic mice correlated with the duration of prevascularization and the number of implanted islets in the TEC site

Because a highly developed microvascularized network had been established in the TEC on day 28 post-implantation, we hypothesized that the normalization of blood glucose levels in diabetic mice was correlated with the duration of prevascularization and the number of islets implanted. To prove this hypothesis, 300 syngeneic islets were transplanted into individual TECs on days 0, 14, and 28 post-TEC implantation. As shown in Fig 3A, the sham group did not exhibit a reversal of diabetes at any time point. However, the islets transplanted on day 28 post-TEC implantation reversed diabetes by 7.4±3.6 d. Conversely, islets transplanted on days 0 and 14 post-TEC implantation resulted in a reversal of diabetes after 32.3±11.7 d and 13.0 ±4.2 d, respectively. Thus, the results suggest that normalization of blood glucose levels in diabetic mice was negatively correlated with the duration of prevascularization.

The effect of the number of islets on blood glucose level normalization in diabetic mice after prevascularization was further investigated. A total of 300 (n = 5), 200 (n = 5), and 100 (n = 5) islets were transplanted on day 28 post-implantation, and the time required for blood glucose to recover was determined. As shown in Fig 3B, 300 islets were more effective at reversing hyperglycemia, which was achieved after 7.4±3.6 d, than were 200 (14.0±3.5 d) or 100 islets (20.4±4.7 d). This indicates that the return to normal levels of blood glucose in the diabetic mice was negatively correlated with the number of islets transplanted.

The long-term function of syngeneic islets within a TEC was then evaluated. Syngeneic islets transplanted into a TEC reversed diabetes in 100% of recipients over the long term (5/5) (Fig 3C). Patients with insulin-dependent diabetes mellitus also need to pay attention to weight

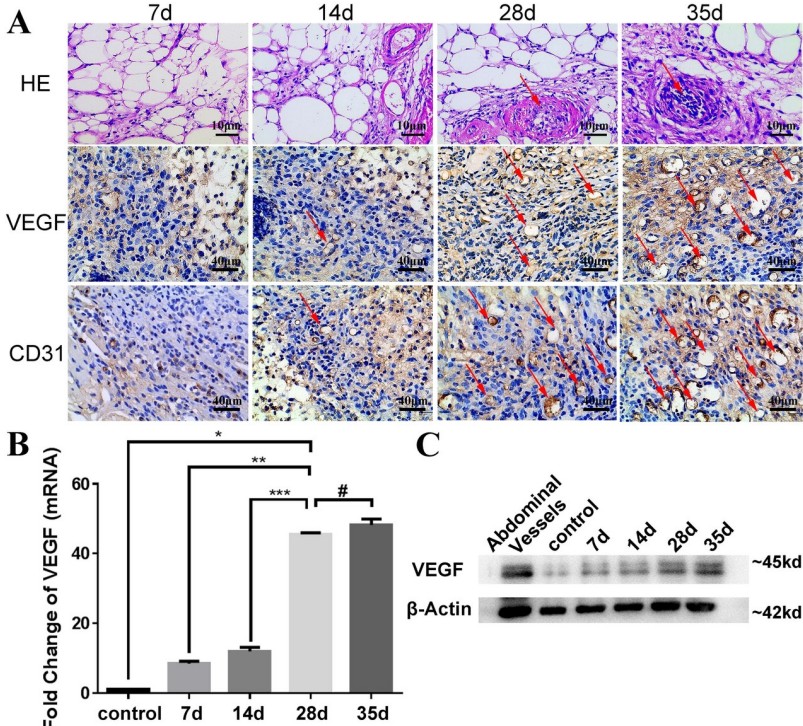

**Fig 2. Highly developed microvascularized network established in the TEC on day 28 post-implantation.** (A) HE images (1st row; scale bars: 10 μm) and histological analysis (2nd and 3rd rows; scale bars: 40 μm) of a prevascularized TEC on days 7, 14, 28 and 35. Red arrows indicate blood vessels in HE staining (1st row; scale bars: 10 μm). Microvessels were stained with anti-VEGF and anti-CD31, as marked separately by red arrows (2nd and 3rd rows; scale bars: 40 μm). (B) Levels of VEGF mRNA expression were quantified. The extent of prevascularization confirmed a significant increase at 28 d compared with 7 d and 14 d ($p^{**} < 0.001$, $p^{***} < 0.001$) and showed no significant difference between the 28 d and 35 d groups ($p^{\#} = 0.0538$). (C) Western blot analysis indicated VEGF protein expression levels at different time points in TECs. The control group had a chamber (but had no Matrigel™ and no heparin), and the sample came from the fat pad tissue near the superficial epigastric vessel. Data are shown as the mean ± SD and are representative of three separate experiments.

in the clinic. Therefore, weight monitoring was conducted in mice both before and after transplantation. The results demonstrate that the body weight of diabetic mice increased gradually when blood glucose was maintained at normal levels (S1A Fig). In addition, histological analysis of grafts revealed that syngeneic islets stained strongly positive for insulin (Fig 3D). These images verified that the TEC is a suitable site for transplantation in future studies.

## Feasibility of the TEC as a transplantation site for long-term survival of allografts and xenografts established by anti-CD45RB and anti-MR-1

Because implantation of allografts and xenografts is a potentially promising solution for insulin-dependent diabetes mellitus, short-term immunosuppression to protect the long-term survival of allografts and xenografts within a TEC was investigated. We first explored whether anti-CD45RB or/and MR-1 therapy could prolong the survival of allografts from recipient C57 mice with STZ-induced diabetes. Three hundred allogeneic islets from 2 donors were transplanted into the TEC, and the nonfasting blood glucose level was monitored. Not surprisingly, treatment with anti-CD45RB or anti-MR-1 alone failed to control hyperglycemia in any recipient (n = 5) (Fig 4A). Conversely, long-term graft survivors (LTS) that were recipients of allografts survived >90 d following combined antibody treatment with anti-CD45RB and anti-

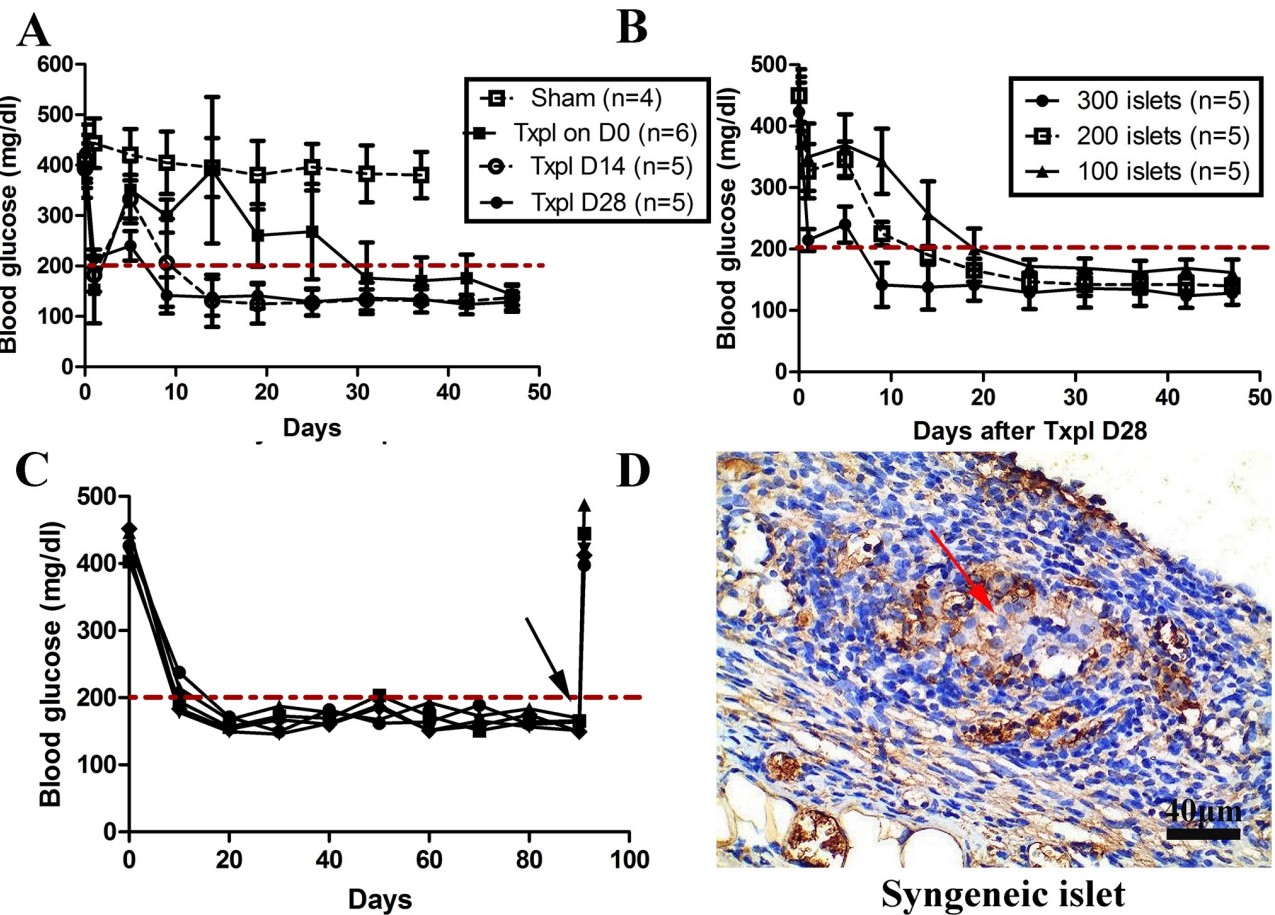

**Fig 3. The TEC represents a promising site for syngeneic islet transplantation.** The time required for blood glucose to return to normal levels (200 mg/dL) in diabetic mice was negatively correlated with the duration of prevascularization (A) and the quantity of seeded syngeneic islets (B). (C) Nonfasting blood glucose showed maintenance of euglycemia after 300 syngeneic islets were transplanted into the TEC, with hyperglycemia developing following TEC removal (black arrow). (D) Donor islets stained positive for insulin in a recipient TEC 90 d post-transplantation (shown brown), indicated by red arrow (Scale bars: 40 μm). Diabetic mice that had neither constructed chambers nor transplanted islets served as the sham group. Txpl: Transplantation.

MR-1 (Fig 4B). Groin grafts containing islets were removed at 90 d post-transplantation, and blood glucose levels rapidly exceeded 400 mg/dL (Fig 4B, black arrow). This indicates that the maintenance of euglycemia in recipient mice was dependent on the islets that were transplanted into the TEC. Histology of the grafts confirmed that implanted islets stained strongly positive for insulin (Fig 4F, left).

Whether xenografts exhibited a similar function in the TECs was then examined. A total of 300 xenogeneic islets were implanted to explore their long-term performance in recipient C57 mice with STZ-induced diabetes. As shown in Fig 4C, treatment with anti-CD45RB or anti-MR-1 alone failed to control hyperglycemia in any xenograft recipient (n = 5). However, sixteen of the twenty recipients maintained euglycemia following immunotherapy (Fig 4C and 4D). We reimplanted xenogeneic islets into mice in which the reversal of hyperglycemia had failed and explored whether euglycemia could be maintained over the long term following dual anti-CD45RB plus anti-MR-1 antibody treatment. The results revealed that three of the four recipient mice experienced a reversal of diabetes and maintained euglycemia over the long term (Fig 4E).

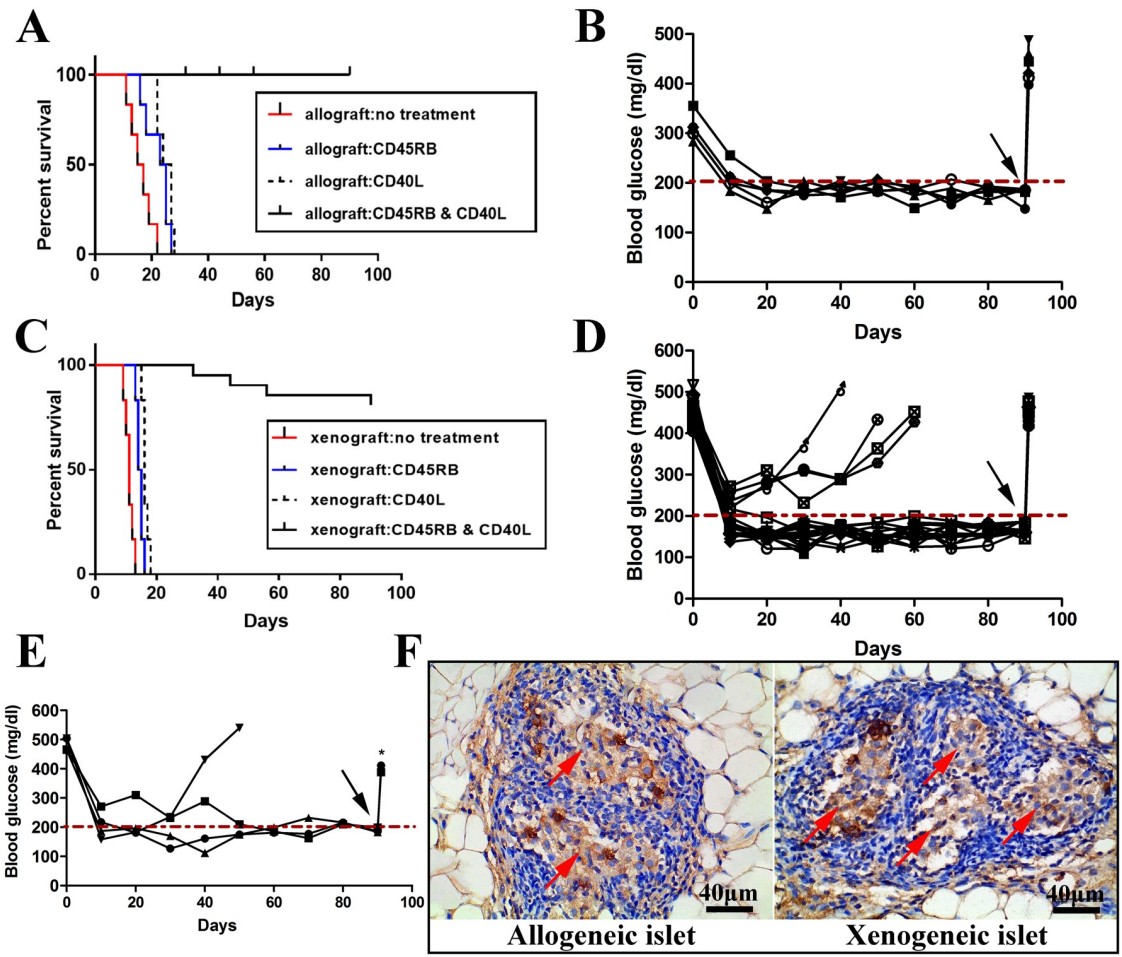

**Fig 4. Long-term function of allogeneic and xenogeneic islet grafts transplanted into TECs.** (A) Islet allografts (Balb/c to C57BL/6) in the no treatment group were rejected within 22±2.2 d. Islet allografts treated with anti-CD45RB or anti-MR-1 alone were also rejected in recipients within 24±3.12 d or 26±1.74 d, respectively. (B) Islet allografts subjected to dual anti-CD45RB or anti-MR-1 treatment displayed long-term graft survival in all recipients (5/5, 100%), which became hyperglycemic following TEC removal (black arrow). (C) Islet xenografts (SD rat to C57BL/6 mouse) were rejected in the no treatment group, the CD45RB-treated group and the MR-1-treated group after 13±3.1 d, 16±3.12 d, and 18±1.74 d, respectively. Islet xenografts treated with dual anti-CD45RB and anti-MR-1 exhibited long-term graft survival (16/20, 80%) and recurrence of hyperglycemia following TEC removal (black arrows). (E) A total of 300 xenogeneic islets were retransplanted into diabetic mice that had failed to accept the xenograft. In three of four diabetic mice, hyperglycemia was ameliorated until the TEC was removed (black arrow). (F) Insulin-stained donor allografts and xenografts in a recipient TEC 90 d post-transplantation (shown in brown). Red arrows indicate islets (Scale bars: 40 μm).

Additionally, the body weight of diabetic mice in which allografts and xenografts had been implanted increased gradually when blood glucose was maintained at normal levels (S1B and S1C Fig). The allogeneic and xenogeneic islets implanted within TECs displayed intense positive staining for insulin (Fig 4F, right). After 90 d, an IPGTT was conducted. Both the allograft and xenograft groups had well-preserved glucose clearance profiles similar to those of the naive group (S2A Fig). These data reveal that allografts and xenografts implanted into TECs could maintain euglycemia over the long term.

## Xenogeneic islets located in TECs induced long-term recipient-specific immune tolerance

The evidence suggests that the persistence of memory T cells (CD4 and CD8) after transplantation may lead to rejection, which destroys most insulin-producing β cells [25]. Therefore,

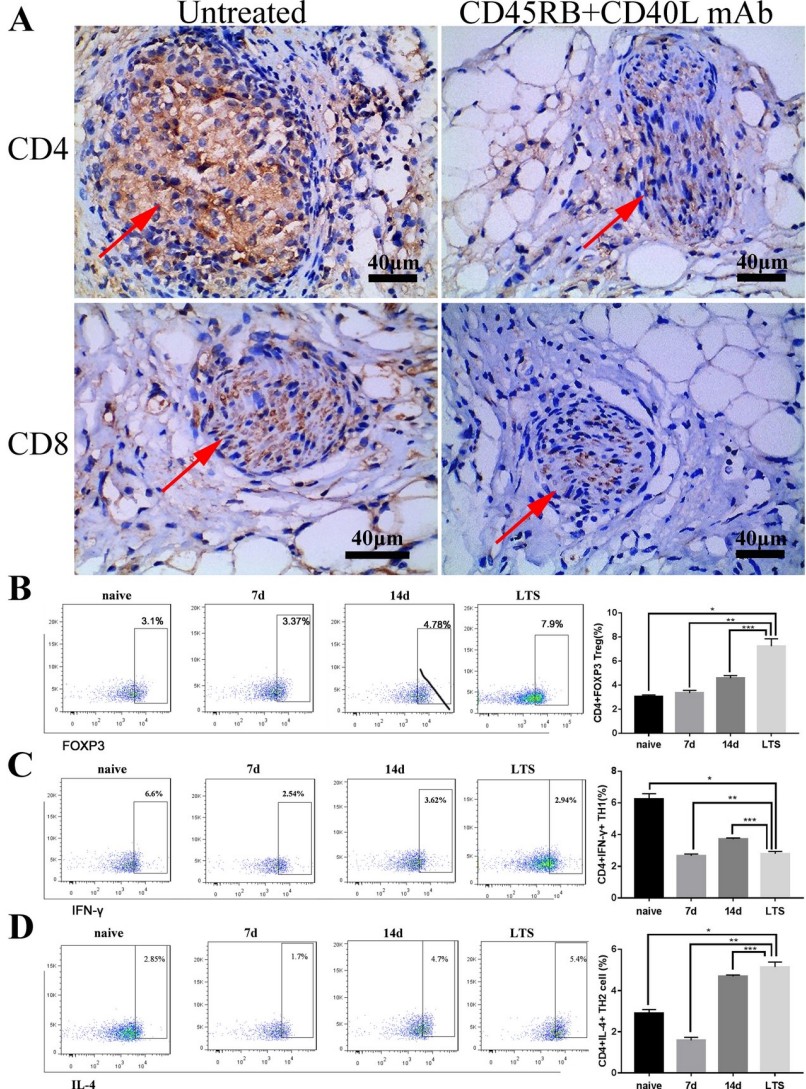

**Fig 5. Anti-CD45RB and anti-MR-1 induced immune tolerance in TECs.** (A) CD4+ and CD8+ T lymphocytes were evaluated by immunohistochemistry after islet transplantation. The group with no treatment (right) demonstrated significant infiltration (brown region) compared with that of the group treated with anti-CD45RB and MR-1 (left). Red arrows indicate islets. Scale bars: 40 μm; The proportions of CD4+ FOXP3+ Treg, CD4+ IFN-γ+ Th1 and CD4 + IL-4+ Th2 cells were measured by FACS after transplantation for 7, 14 and 90 d (LTS) (n = 3 mice per group). (B) The percentage of Tregs in the LTS group was significantly higher than those in the 7 d and 14 d groups. (C) Numbers of CD4+ IFN-γ+ Th1 cells decreased in the 7-day and LTS groups. (D) The number of CD4+ IL-4+ Th2 cells increased significantly on days 14 and 90. FACS histograms are representative of at least three independent experiments examined on days 7, 14, 90 after transplantation ($p^* < 0.05$ versus naive, $p^{**} < 0.005$ vs. 7 d group, $p^{***} < 0.005$ vs. 14 d group).

inflammatory infiltration was evaluated by immunohistochemistry in xenografts on the 90th day after transplantation. Infiltration of CD4+ and CD8+ T cells decreased significantly after immunotherapy (anti-CD45RB plus MR-1) in comparison with the untreated group (Fig 5A). These results showed that xenogeneic islets transplanted in TECs were protected during immunotherapy.

T regulatory cells (Tregs) play a central role in maintaining immune homeostasis and peripheral tolerance to foreign antigens in the body. Therefore, it is important to explore the

development of immune tolerance in response to variations in the numbers of Treg cells in xenograft recipients. The numbers of CD4[+] Foxp3[+] Tregs were measured at different time points after transplantation, including 7 d, 14 d, and 90 d (LTS), using flow cytometry. As displayed in Fig 5B, the percentage of Tregs in the LTS group increased significantly compared with that in the 7 d and 14 d groups. The proportion of CD4[+] Foxp3[+] Treg cells increased from a normal value of 3.1% to almost 7.9% on day 90 after transplantation. In addition, we attempted to gain insight into the distribution of functional helper T cell subsets in mice to illustrate the development of immune tolerance. Splenocytes were stimulated for 6 hours using a cell stimulation cocktail prior to staining for intracellular markers. Compared with those in the LTS group, the levels of IFN-γ-producing CD4[+] T cells (Th1) were lower in the 7 d and 14 d groups (Fig 5C). The number of IL-4-producing CD4[+] T (Th2) cells in the LTS group was higher than those in the 7 d and 14 d groups. A significant difference was observed between the LTS group and other groups (Fig 5D). Changes in the Th1/Th2 ratio, further indicating the T cell response, were skewed toward tolerance.

## Discussion

In the present study, a silicone tube chamber with a rich blood vessel network was constructed using a tissue engineering method. The feasibility and safety of transplantation were considered in detail. Silicone is a safe material that induces a minimal tissue response in animals and humans [26]. The present study evaluated a 5 mm-long cylindrical silicone chamber filled with Matrigel™ matrix placed in the groins of mice surrounding the superficial epigastric vessel. Matrigel™ originated from the Engelbreth-Holm-Swarm mouse sarcoma and was found to be a substrate that induces angiogenesis in vivo [27]. In particular, major components, including laminin and collagen IV, modulate cell signaling interactions and are critical for pancreatic β-cell survival and differentiation [28]. A number of studies have reported that neovascularization can be observed as early as 2 weeks, with a network of vessels appearing in the islets by day 21 [23]. In our study, the results revealed that an abundant microvascularized system was established by day 28 in the TEC. Furthermore, following the transplantation of 300 syngeneic islets into TECs in diabetic mice, it is easy to observe that the blood glucose of the diabetic mice was restored to euglycemia after 7.4±3.6 d. The results are similar to those obtained with islets seeded into a kidney capsule in diabetic mice, which is the most common method of transplantation in basic research [13, 29]. Some studies have demonstrated that a microenvironment created from the integration of a vascular network could improve the survival of grafts in the long term [30]. In addition, the abundant prevascularization in the TEC supplies more nutrition to the islets, supporting enhanced survival and function compared with the transplantation site of the kidney capsule. However, we have yet to study autoimmune diabetes models, such as NOD or NRG Akita mice, which exhibit impaired vascularity associated with human diabetes [31]. It is essential to understand the relationship between islet viability within a TEC and revascularization in the future.

Intrahepatic transplantation not only activates an inflammatory reaction in the blood but also fails to support the survival of implanted islets [32]. Therefore, the liver can be considered a suboptimal site for islet transplantation. Many other sites have been reported, including the greater omentum [33, 34], kidney [35], spleen [36], gastric submucosa [37], epididymal fat [38], muscle tissue [39], and anterior chamber of the eye [40], and the level of blood glucose has been demonstrated to be controlled. Nevertheless, pancreatic islet survival and function in these locations remain unsatisfactory [41] due to invasive surgery, inadequate angiogenesis, and the corresponding lack of direct integration with the host. In our study, the TEC chamber

was designed and placed in a subcutaneous site as the first step prior to subsequent islet transplantation in the second step. The strategies were effective in achieving prolonged glycemic control over the long term. In addition, the grafts were implanted, monitored, and removed quickly if local complications such as malignant transformation or unchecked hormone release were observed. In summary, subcutaneous sites can be regarded as an alternative for the transplantation of pancreatic islets [42].

Transplantation of pancreatic islets is a promising treatment for insulin-dependent diabetes mellitus. However, the limited availability of pancreatic islets limits their clinical use. In theory, islets derived from animals (porcine islets) can solve the problem of organ shortage, but problems associated with strong rejection need to be solved in the future [43]. The mechanism of rejection following xenogeneic transplantation is extraordinarily complicated, as shown by the small number of reports of successful induction of xenografts [44, 45]. In the present study, euglycemia was maintained in the TEC by an effective dual antibody treatment (anti-CD45RB plus anti-MR-1) after xenotransplantation from rats to C57 mice. CD45 and CD40 are both TCR surface receptors [46, 47]. The use of these two antibodies can significantly inhibit the proliferation of T cells. As shown in Fig 5A, dual antibody treatment prolonged islet survival and protected them from destruction by an inflammatory response, maintaining their initial shape, structure, and function of insulin secretion. In addition, the protective potential of $CD4^+$ $Foxp3^+$ Treg and $CD4^+$ $IL-4^+$ T cells in xenotransplantation has been identified previously [48, 49]. According to the findings of the present study, we believe that anti-CD45RB plus MR-1 significantly inhibits the cellular immune response in TECs, which decreases the proportion of $CD4^+$ $IFN-\gamma^+$ Th1 cells and increases the numbers of $CD4^+$ $IL-4^+$ Th2 cells and $CD4^+$ $Foxp3^+$ Treg cells for long-term tolerance.

Remarkably, islets transplanted into the TEC formed a pancreatic "organoid". This construct remained intact and removable. The retrievability of the construct is an important clinical consideration for future work because it allows retransplantation or the insertion of alternative insulin-producing cells, such as stem cell-derived pancreatic progenitor cells, following complications. These results suggest that the TEC is a promising strategy for clinical application in islet transplantation.

In summary, the present study demonstrated the safety and feasibility of the TEC as a promising site that promotes the survival of islets. Tissue-engineered prevascularized chambers can be expected to become an attractive therapy for insulin-dependent diabetes mellitus.

## Supporting information

**S1 Fig. Variation in weight from TEC implantation to islet transplantation.** Continued weight stability indicated the overall safety of the TEC 28 d post-implantation. Weight increased after transplantation, closely correlating with treatment regimens in the TEC. (TIF)

**S2 Fig. Comparison of IPGTTs of syngeneic, allogeneic, and xenogeneic islets 90 days after transplantation in a TEC.** The naive mice were nondiabetic, nontransplanted C57 mice (black, $n = 3$), which are more tolerant of metabolic tests than transplant recipients. Blood glucose measurements were monitored at $t = 0, 15, 30, 60, 90,$ and 120 minutes. Data points represent the mean ± S.E.M. of blood glucose values. No difference in the tolerance of mice to glucose challenge was observed in mice that received syngeneic, allogeneic, or xenogeneic islets in a TEC ($n = 3$) compared to naive animals ($p^* > 0.01$ vs. syngeneic group, $p^{**} > 0.01$ vs. allogeneic group, $p^{***} > 0.01$ vs. xenogeneic group). (TIF)

**S1 Raw data.**
(PDF)

## Author Contributions

**Data curation:** Yanzhuo Liu, Maozhu Yang.

**Formal analysis:** Yanzhuo Liu.

**Investigation:** Yanzhuo Liu, Hao Yuan.

**Methodology:** Yanzhuo Liu, Minxue Liao, Hao Yuan.

**Supervision:** Yuanyuan Cui, Yuanyuan Yao.

**Visualization:** Yanzhuo Liu, Minxue Liao, Guojin Gong.

**Writing – original draft:** Yanzhuo Liu, Yuanyuan Cui.

**Writing – review & editing:** Yanzhuo Liu, Maozhu Yang, Shaoping Deng, Gaoping Zhao.

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
