## [Decision Letter · Decision Letter 0]

22 Jun 2020

PONE-D-20-16218

A novel pre-vascularized tissue engineered chamber as a site for allogenic and xenogeneic islet transplantation to establish a bioartificial pancreas

PLOS ONE

Dear Dr. Zhao,

Thank you for submitting your manuscript to PLOS ONE. After careful consideration, we feel that it has merit but does not fully meet PLOS ONE’s publication criteria as it currently stands. Therefore, we invite you to submit a revised version of the manuscript that addresses the points raised during the review process.

Please answer the comments of especially reviewer #2 very carefully, as he/she had several important technical concerns - these have to be addressed  - both reviewers noted a lack of novelty, but I believe that confirming concepts is indeed very important in science and PlosOne is appropriate for this.

We look forward to receiving your revised manuscript.

Kind regards,

Matthias G von Herrath, MD PhD

Academic Editor

PLOS ONE

Journal Requirements:

2. To comply with PLOS ONE submissions requirements, please provide methods of sacrifice in the Methods section of your manuscript.

"This study was supported by grants from the National Natural Science

Foundation of China (nos. 81172832 and 81771723), Sichuan Youth

Science and Technology Foundation (no. 2013JQ0020), and Special

Program for Sichuan Youth Science and Technology Innovation (no.

2014TD0010)."

" The funders had no role in study design, data collection and analysis, decision to publish, or preparation of the manuscript."

Reviewers' comments:

Reviewer's Responses to Questions

**Comments to the Author**

1. Is the manuscript technically sound, and do the data support the conclusions?

Reviewer #1: Yes

Reviewer #2: No

2. Has the statistical analysis been performed appropriately and rigorously? 

Reviewer #1: Yes

Reviewer #2: No

3. Have the authors made all data underlying the findings in their manuscript fully available?

Reviewer #1: Yes

Reviewer #2: Yes

4. Is the manuscript presented in an intelligible fashion and written in standard English?

Reviewer #1: Yes

Reviewer #2: No

5. Review Comments to the Author

Reviewer #1: This is in principle an honest and straightforward study on implantation of a longitudinally cut silicone tube filled with matrigel to induce pre-vascularization. Subsequently, islets from autologous, allo or xeno origin are then transplanted and outcome monitored. It was found that the longer pre-vascularization went on, the better outcomes tended to be. Since the device is essentially open, both allo and xeno islets required immune suppression (here by anti-CD45 and anti-CD40L). Finally, in these longterm surviving islet recipients, some evidence of tolerized immunity was observed.

The problem is not data quality or clarity of the report but rather my impression of lack of novelty. Did we know vascularization matters greatly to survival? Yes. Will an open device like this accommodate protection against alloreactivity? No. And then the clinical question is whether immune suppression as in this paper is acceptable in the future. Likely not. So the paper fails to show an improved safety efficacy-perspective in my mind.

Minor comment: the authors state that 'the principle concern in clinic is weight'. I see more problems that that so best omit this statement.

Reviewer #2: The introduction summarizes well the shortcomings of intra-hepatic islet infusions and captures nicely the main challenges of extra-hepatic islet transplantation, including tissue capacity, kinetics, and adequate oxygen and nutrient delivery to the graft.

The paper hypothesises that the ability and timing of transplanted islets to normalize blood glucose in diabetic mice correlates to the degree of pre-vascularization of the tx site and the number of transplanted islets, respectively. By employing a tissue engineering technique developed in the late 80’s and combing with Matrigel, the authors examine the degree of vascularisation in the tissue engineered chamber (TEC) 7d, 14d, 28d and 35d after implantation and the effect on blood glucose when 300 syngeneic islets are transplanted into the TEC after 0, 14 and 28 days of pre-vascularization of the TEC. The effect of islet numbers on blood glucose lowering is examined by transplanting 100, 200 and 300 syngeneic islets after 28 days of pre-vascularisation. The authors examine if allo- and xenogeneic islets can survive and function for prolonged time in the TEC by treating the diabetic mice with anti-CD45RB treatment for 7 days +/- anti-CD40L for up to 14 days.

The authors show, that a 28d pre-vascularised TEC is permissive for long term (90d) islet function and survival in syngeneic transplantations and that increasing islet numbers decreases the time interval from islet injection to blood glucose normalisation. Furthermore, the authors provide data indicating, that combination treatment with anti-CD45RB (7d) + anti-CD40L (14d) is permissive for allo-and xenogeneic islet transplantations in this model.

The observed effect of pre-vascularization is in accordance with numerous studies on the effect of oxygen on islet survival and function in vivo and the protective effect of anti-CD45RB (7d) and anti-CD40L has also been demonstrated previously.

The introduction section is relatively well written and addresses relevant background information.

The result section, however, appears to have been written in haste, with little proof-reading. Graphs are mixed up, legends flawed, there are incomplete references and both linguistic and argumentative flaws. The discussion section appears written in haste as well, with several errors and lacks perspective.

I have listed some of the issues below, but the list is by no means exhaustive. I recommend the authors to revise the language to improve readability and correct the flaws before the manuscript can be properly reviewed. I have therefore chosen not to follow the otherwise helpful outline of how to review, since it does not make sense at this stage in my opinion. In its current form, I am inclined to reject the article.

Issues to clarify/correct:

1. The authors refer to allo- or xeno islet transplantation as a potentially promising solution for insulin-independent diabetes mellitus both in the abstract and on p 16, p 21. Do the authors mean insulin-dependent?

2. On page 16, the authors write that they explored whether anti-CD45RB (7d) and anti-CD40L could prolong the survival of the mice. Do they mean survival of the graft?

3. Page 16 bottom, the authors state that the data in Fig 4B indicates that glucose tolerance was dependant on the number of transplanted islets. This does not make sense in that context. Please correct.

4. On page 20-21, the paper lists several sites that have “never been used clinically”, including omentum, muscle and anterior chamber of the eye. It the authors mean to state, that these sites have never been tested clinically, this is incorrect.

5. Fig 3: Fig 3A and 3B has been switched. Thus, the text and the figure legend refer to the wrong graph. The x-axis legend on current fig 3B must be corrected (not 28d). In Fig 3C, please state the number of islets (300?)

6. P15, referral to suppl. Fig 1 for bodyweight. However, this is Suppl. Fig 2. Please correct.

7. Please double check references; the method of creating the TEC is referred to 23 and 24, Hussey et al. and Menger et al. but the method is only described in Cronin et al. which is mentioned in the text but not on the reference list.

8. Fig 4: in the legends of the graphs (red line), what is meant by “uncontrol”?. In the text legend to fig 4, how can 5/5=80%?

9. Fig 3: The 5 animals in the group given 300 islets on d 28 are the same in the two graphs (A & B). Were these two experiments run simultaneously?

10. Since the paper concludes that the TEC is a promising strategy for islet transplantation due to the vascularisation induced by Matrigel, it would be of interest to hear the authors view on the usability of Matrigel in the clinic and the scalability of this technology?

11. The method of creating the TEC and transplanting the islets is unclear and should be clarified, since the referenced papers do not. Are the islets inserted into the tube through the fat pad/ bone wax used to seal the tube? Does this require reopening of the surgical incision made to insert the silicone tube?

12. Histology: Please check if 5mm thick sections are correct. Also, please clarify the spatial origin of the tissue shown in Fig 2: is this from the periphery, the centre or the end of the tube? Is the vascularisation evenly distributed throughout the TEC? Please show HE stained sections from the tissue surrounding the TEC to support the statement that the silicone is well tolerated. Are the images in row 1, 2 and 3 in Fig 2 from the same/adjacent sections?

13. Fig 2. Please clarify if sham is control silicone tube with no Matrigel and no Heparin as in Fig 3. If it is inguinal adipose tissue with no tube, please show sections from the no Matrigel control.

14. The inner diameter of the silicone tube is said to be 3.7 mm. Please show histology of islets in the centre of the tube/tissue chamber.1.85 mm is a long diffusion distance for oxygen to reach the central tissue and may significantly impact the capacity of this technology by only allowing survival of islets in the periphery of the tube.

15. In the discussion, the paper mentions that grafts were removed if teratoma or malignant transformation was observed. From where did teratomas arise? Adult islets do not contain pluripotent cells. Does the Matrigel induce uncontrolled growth of the islets or native cells? Please discuss how this affects the safety and relevance of this model.

16. It would be of interest if the authors would discuss the scalability of this approach to clinically relevant numbers of islets (in the range of 400K-800K IEQs based on portal vein injections in 60-80 kg patients)

17. It is concerning, that only 5 animals are included in each group, and I am surprised to see so little variation, since there normally is significant variation among animals. Raw data should be reviewed. Statistical analysis of samples from 5 animals is not sufficiently robust to draw solid conclusions from. Has power analysis been performed?

6. PLOS authors have the option to publish the peer review history of their article (what does this mean?). If published, this will include your full peer review and any attached files.

Reviewer #1: No

Reviewer #2: No

---

## [Author Response · Author response to Decision Letter 0]

21 Sep 2020

Dear editorial-lmanager and Reviewers,

Thank you for providing the valuable suggestions and comments for our manuscript (ID: PONE-D-20-16218). We have carefully revised the manuscript according to the reviewer comments. The answers to the comments are listed as following.

Reply to Reviewer #1

Comment. This is in principle an honest and straightforward study on implantation of a longitudinally cut silicone tube filled with matrigel to induce pre-vascularization. Subsequently, islets from autologous, allo or xeno origin are then transplanted and outcome monitored. It was found that the longer pre-vascularization went on, the better outcomes tended to be. Since the device is essentially open, both allo and xeno islets required immune suppression (here by anti-CD45 and anti-CD40L). Finally, in these long-term surviving islet recipients, some evidence of tolerized immunity was observed. 

Comment 1. The problem is not data quality or clarity of the report but rather my impression of lack of novelty. 

1. Answer. In many study, various strategies were designed to per-vascularization before islet transplantation[1]. However, we first proposed the feasibility about the allogeneic and xenogeneic islet transplantation in the subcutaneous. If the rejection occurs after transplantation, we can still re-transplanted islets in the TECs. Therefore, the TECs can be designed as an artificial pancreas organ and seeded fresh islet repeatedly. In theory, this study is important for development biomaterials on xenotransplantation. Meanwhile, this principle is very meaningful for exploration artificial pancreas organ.

Comment 2. Did we know vascularization matters greatly to survival? Yes. Will an open device like this accommodate protection against alloreactivity? No. 

2. Answer. Based on our knowledge, the encapsulation of pancreatic islets allows for transplantation in the absence of immunosuppression. The technology is based on the principle that transplanted tissue is protected for the host immune system by an artificial membrane[2]. The TECs alone cannot avoid rejection in our study, so we used immunosuppressive agents (anti-CD45RB and anti-CD40L) to protection against alloreactivity. The CD45 and CD40 are both TCR surface receptors [3, 4]. The use of these two antibodies can significantly inhibited the proliferation of T cell and the occurrence of immune rejection.

Comment 3. And then the clinical question is whether immune suppression as in this paper is acceptable in the future. Likely not. 

3. Answer. I agree with you, the immunologic therapy (anti-CD45RB and anti-CD40L) in the article have not applied in the clinic. Although, these two antibodies have potent resistance to immune rejection in animal studies[5], there is still a lack of feasibility and safety research in large animals. Therefore, the application of anti-CD45RB and anti-CD40L is a little far away in the clinic.

Comment 4. So the paper fails to show an improved safety efficacy-perspective in my mind. Minor comment: the authors state that 'the principle concern in clinic is weight'. I see more problems that that so best omit this statement.

4. Answer. In our study, we evaluated the safety in construction of TECs and immunologic therapy. We discovered that none of the mice suffered leg paralysis, wound infection, chamber exposure, or sinus after the pre-vascularization chamber was constructed. It shows that the TECs has good safety and biocompatibility. Subsequently the normalization of blood glucose levels in the diabetic mice when allogeneic and xenogeneic islets transplanted into the TECs, and the weight also showed an upward trend. Histology of the grafts (>90d) showed that islets stained strongly-positive for insulin. I have deleted the statement which are incorrect in the study.

Reply to Reviewer #2

Comment. The introduction summarizes well the shortcomings of intra-hepatic islet infusions and captures nicely the main challenges of extra-hepatic islet transplantation, including tissue capacity, kinetics, and adequate oxygen and nutrient delivery to the graft.The paper hypothesises that the ability and timing of transplanted islets to normalize blood glucose in diabetic mice correlates to the degree of pre-vascularization of the tx site and the number of transplanted islets, respectively. By employing a tissue engineering technique developed in the late 80’s and combing with Matrigel, the authors examine the degree of vascularisation in the tissue engineered chamber (TEC) 7d, 14d, 28d and 35d after implantation and the effect on blood glucose when 300 syngeneic islets are transplanted into the TEC after 0, 14 and 28 days of pre-vascularization of the TEC. The effect of islet numbers on blood glucose lowering is examined by transplanting 100, 200 and 300 syngeneic islets after 28 days of pre-vascularisation. The authors examine if allo- and xenogeneic islets can survive and function for prolonged time in the TEC by treating the diabetic mice with anti-CD45RB treatment for 7 days +/- anti-CD40L for up to 14 days. The authors show, that a 28d pre-vascularised TEC is permissive for long term (90d) islet function and survival in syngeneic transplantations and that increasing islet numbers decreases the time interval from islet injection to blood glucose normalisation. Furthermore, the authors provide data indicating, that combination treatment with anti-CD45RB (7d) + anti-CD40L (14d) is permissive for allo-and xenogeneic islet transplantations in this model. The observed effect of pre-vascularization is in accordance with numerous studies on the effect of oxygen on islet survival and function in vivo and the protective effect of anti-CD45RB (7d) and anti-CD40L has also been demonstrated previously. The introduction section is relatively well written and addresses relevant background information. The result section, however, appears to have been written in haste, with little proof-reading. Graphs are mixed up, legends flawed, there are incomplete references and both linguistic and argumentative flaws. The discussion section appears written in haste as well, with several errors and lacks perspective. I have listed some of the issues below, but the list is by no means exhaustive. I recommend the authors to revise the language to improve readability and correct the flaws before the manuscript can be properly reviewed. I have therefore chosen not to follow the otherwise helpful outline of how to review, since it does not make sense at this stage in my opinion. In its current form, I am inclined to reject the article.

Comment 1. The authors refer to allo- or xeno islet transplantation as a potentially promising solution for insulin-independent diabetes mellitus both in the abstract and on p 16, p 21. Do the authors mean insulin-dependent?

1. Answer. yes, it means the “insulin-dependent diabetes mellitus”. I have corrected the errors of typing in the original text.

Comment 2. On page 16, the authors write that they explored whether anti-CD45RB (7d) and anti-CD40L could prolong the survival of the mice. Do they mean survival of the graft?

2. Answer. Immunologic therapy (anti-CD45RB and anti-CD40L) could prolong the survival of the graft. We have revised it in the original text. 

Comment 3. Page 16 bottom, the authors state that the data in Fig 4B indicates that glucose tolerance was dependent on the number of transplanted islets. This does not make sense in that context. Please correct.

3. Answer. We have been revised in the original text. 

Comment 4. On page 20-21, the paper lists several sites that have “never been used clinically”, including omentum, muscle and anterior chamber of the eye. It the authors mean to state, that these sites have never been tested clinically, this is incorrect.

4. Answer. Other studies have also explored seeding islets into different locations, including omentum, muscle and anterior chamber of the eye, which could maintain the normal level of blood glucose in the clinic. However, pancreatic islet survival and function in these locations remain unsatisfactory[6, 7]. The expression about “never been used clinically” is incorrectand we have been revised in the original text. 

Comment 5. Fig 3: Fig 3A and 3B has been switched. Thus, the text and the figure legend refer to the wrong graph. The x-axis legend on current fig 3B must be corrected (not 28d). In Fig 3C, please state the number of islets (300?)

5. Answer. We have been revised in the original text. 

Comment 6. P15, referral to suppl. Fig 1 for bodyweight. However, this is Suppl. Fig 2. Please correct.

6. Answer. The original text has been revised.

Comment 7. Please double check references; the method of creating the TEC is referred to 23 and 24, Hussey et al. and Menger et al. but the method is only described in Cronin et al. which is mentioned in the text but not on the reference list.

7. Answer. The original text has been revised.

Comment 8. Fig 4: in the legends of the graphs (red line), what is meant by “uncontrol”?. In the text legend to fig 4, how can 5/5=80%?

8. Answer. Un-control is the group of no treatment by anti-CD45RB or/and anti-CD40L. The description of “unconrtol” was incorrectly stated in the original manuscript. We have been corrected it. 

Comment 9. Fig 3: The 5 animals in the group given 300 islets on d 28 are the same in the two graphs (A & B). Were these two experiments run simultaneously?

9. Answer. The group of Txpl D28 in Fig3 A and the group of 300 islets in Fig B is same experiment. In the study, we first transplanted 300 islets into the TECs on days 0, 14 and 28 post-chamber implantation. In order to explore the relationship between the time of pre-vascularization and blood glucose. Subsequently, 100, 200, and 300 islets were injected into the TECs on 28 day of pre-vascularization. In order to explore the relationship between the number of islets and blood glucose. 

Comment 10. Since the paper concludes that the TEC is a promising strategy for islet transplantation due to the vascularisation induced by Matrigel, it would be of interest to hear the authors view on the usability of Matrigel in the clinic and the scalability of this technology?

10. Answer. MatrigelTM is used for lab research only. It originated from the Engelbreth-Holm-Swarm mouse sarcoma and has been found to be a substrate that induces angiogenesis in vivo. In particular, major components, including laminin and collagen IV, modulate cell signaling interactions, is critical for pancreatic β-cell survival and differentiation. Extracellular matrix (ECM) bio-scaffolds (similar to materigelTM) prepared from decellularized tissues have been used to facilitate constructive and functional tissue remodeling in a variety of clinical applications.[8, 9]. The matrix may be modified to induce neo-vascularization by biology.

Comment 11. The method of creating the TEC and transplanting the islets is unclear and should be clarified, since the referenced papers do not. Are the islets inserted into the tube through the fat pad/ bone wax used to seal the tube? Does this require reopening of the surgical incision made to insert the silicone tube?

11. Answer. The image about constructing TECs and islet transplantation have been added in the original text. The referenced papers about transplantation is same as the construction of TECs, I have been corrected it in the original text. The surgical incision was reopened where the silicone tube was insert, and the scalp needle containing the islets inserted into the tube through the bone wax.

Comment 12. Histology: Please check if 5mm thick sections are correct. Also, please clarify the spatial origin of the tissue shown in Fig 2: is this from the periphery, the centre or the end of the tube? Is the vascularisation evenly distributed throughout the TEC? Please show HE stained sections from the tissue surrounding the TEC to support the statement that the silicone is well tolerated. Are the images in row 1, 2 and 3 in Fig 2 from the same/adjacent sections?

12. Answer. The slice thickness is 5μm which has been corrected in the original text. The image of HE in figure 2 is from the center which close to the fat pad. The image of immunohistochemistry is from the center of TECs. The method of pre-vascularization is implanted adjacent to the superficial epigastric vessel in the groin. Therefore, the extension of blood vessels is based on the superficial epigastric vessel and the pre-vascularization are not completely evenly distributed. 

We understand that the tissue surrounding of TECs may better reveal the tolerated and bio-compatibility. However, we mainly focused on the feasibility of the TECs of pre-vascularized as a site to islet transplantation in the present study. We observed that none of mice have leg paralysis, wound infection, exposed cavities after TECs was constructed. It should be sufficient to draw a conclusion that the TECs is well bio-compatibility. 

Comment 13. Fig 2. Please clarify if sham is control silicone tube with no Matrigel and no Heparin as in Fig 3. If it is inguinal adipose tissue with no tube, please show sections from the no MatrigelTM control.

13. Answer. In the figure 2B and 2C, the sham group should be the group of control which have a chamber (but have no matrigelTM and no heparin). I have corrected the errors of typing in the original text. In figure3A, the diabetic mice, which have neither constructed chamber nor transplanted islets, were acted as sham group. The sample is derived from the fat pad tissue near the superficial epigastric vessel, the image of HE was showed.

Comment 14. The inner diameter of the silicone tube is said to be 3.7 mm. Please show histology of islets in the centre of the tube/tissue chamber.1.85 mm is a long diffusion distance for oxygen to reach the central tissue and may significantly impact the capacity of this technology by only allowing survival of islets in the periphery of the tube.

14. Answer. The study is to explore the feasibility of islet transplantation by constructing a tissue engineering pre-vascularization chamber in the subcutaneous. Meanwhile, we also consider the factor of oxygen diffusion. The islets are transplanted near the superficial epigastric vessel where have abundant neo-vascularization.

Comment 15. In the discussion, the paper mentions that grafts were removed if teratoma or malignant transformation was observed. From where did teratomas arise? Adult islets do not contain pluripotent cells. Does the Matrigel induce uncontrolled growth of the islets or native cells? Please discuss how this affects the safety and relevance of this model.

15. Answer. The matrigelTM is a safe basement membrane to islets and would not induce mature cell again. Nowadays, stem cells are co-transplanted with islet cells that can improve the outcome of islet transplantation[10]. Therefore, the islets are co-cultured with stem cell which caused teratomas due to over-differentiation. However, stem cells were not involved in our study and the teratomas would not happened. We have revised relevant incorrect information in original text.

Comment 16. It would be of interest if the authors would discuss the scalability of this approach to clinically relevant numbers of islets (in the range of 400K-800K IEQs based on portal vein injections in 60-80 kg patients)

16. Answer. The question is very interesting. There are still many obstacles, including the uncertainty of the location and size of the chamber. Islets are transplanted into the liver via the portal vein that causes a large number of islet apoptosis due to local hypoxia as a result of the limited blood supply. If we find a suitable location to implantation a chamber of vascularization in the clinic, we believe that the number of transplantation islets will not exceed 400k-800k IEQs. Meanwhile, chamber was transplanted in the subcutaneous, which has the advantage of multiple transplantation islets.

Comment 17. It is concerning, that only 5 animals are included in each group, and I am surprised to see so little variation, since there normally is significant variation among animals. Raw data should be reviewed. Statistical analysis of samples from 5 animals is not sufficiently robust to draw solid conclusions from. Has power analysis been performed? 

17. Answer. The fluctuation of blood glucose have similar among the normal mice (100mg/dL-200mg/dL) [11, 12]. Data were analyzed using Graph-Pad Prism software version 5 (Graph-Pad version 5.01), and the process of data analysis is scientific and reasonable. Under the condition of SPF, their cages were housed within a 12-h day/night cycle and with ad libitum access to food and water. Meanwhile, mice were transplanted and treated under the same experimental conditions. Therefore, there is little individual difference between mice. Based on past experience and other relevant study revealed that the statistical analysis of samples from 5 animals is sufficiently robust [13-15]. Furthermore, we focus on the survival of xeno-graft (n=20) to confirm the accuracy of the results. 

The reviewer’s comments are very helpful, and we have revised the previous manuscript accordingly. For more detailed revisions, please refer to the revised manuscript.

Sincerely yours,

Zhao Gaoping

Department of University of Electronic Science and Technology of China

No.4, Jian She Road, chengdu, 610000, China

Email: gzhao@uestc.edu.cn 

Reference

1. Bowers DT, Song W, Wang LH, Ma M. Engineering the vasculature for islet transplantation. Acta biomaterialia. 2019;95:131-51. Epub 2019/05/28. doi: 10.1016/j.actbio.2019.05.051. PubMed PMID: 31128322; PubMed Central PMCID: PMCPMC6824722.

2. de Vos P, Hamel AF, Tatarkiewicz K. Considerations for successful transplantation of encapsulated pancreatic islets. Diabetologia. 2002;45(2):159-73. Epub 2002/04/06. doi: 10.1007/s00125-001-0729-x. PubMed PMID: 11935147.

3. Chang VT, Fernandes RA, Ganzinger KA, Lee SF, Siebold C, McColl J, et al. Initiation of T cell signaling by CD45 segregation at 'close contacts'. Nature immunology. 2016;17(5):574-82. Epub 2016/03/22. doi: 10.1038/ni.3392. PubMed PMID: 26998761; PubMed Central PMCID: PMCPMC4839504.

4. Unutmaz D, Baldoni F, Abrignani S. Human naive T cells activated by cytokines differentiate into a split phenotype with functional features intermediate between naive and memory T cells. International immunology. 1995;7(9):1417-24. Epub 1995/09/01. doi: 10.1093/intimm/7.9.1417. PubMed PMID: 7495749.

5. Molano RD, Pileggi A, Berney T, Poggioli R, Zahr E, Oliver R, et al. Prolonged islet allograft survival in diabetic NOD mice by targeting CD45RB and CD154. Diabetes. 2003;52(4):957-64. Epub 2003/03/29. doi: 10.2337/diabetes.52.4.957. PubMed PMID: 12663467.

6. Cantarelli E, Piemonti L. Alternative transplantation sites for pancreatic islet grafts. Current diabetes reports. 2011;11(5):364-74. Epub 2011/07/27. doi: 10.1007/s11892-011-0216-9. PubMed PMID: 21789599.

7. Shapiro AM, Pokrywczynska M, Ricordi C. Clinical pancreatic islet transplantation. Nature reviews Endocrinology. 2017;13(5):268-77. Epub 2016/11/12. doi: 10.1038/nrendo.2016.178. PubMed PMID: 27834384.

8. Saldin LT, Cramer MC, Velankar SS, White LJ, Badylak SF. Extracellular matrix hydrogels from decellularized tissues: Structure and function. Acta biomaterialia. 2017;49:1-15. Epub 2016/12/05. doi: 10.1016/j.actbio.2016.11.068. PubMed PMID: 27915024; PubMed Central PMCID: PMCPMC5253110.

9. Abbas Y, Carnicer-Lombarte A, Gardner L, Thomas J, Brosens JJ, Moffett A, et al. Tissue stiffness at the human maternal-fetal interface. Human reproduction (Oxford, England). 2019;34(10):1999-2008. Epub 2019/10/04. doi: 10.1093/humrep/dez139. PubMed PMID: 31579915; PubMed Central PMCID: PMCPMC6809602.

10. Oh BJ, Oh SH, Jin SM, Suh S, Bae JC, Park CG, et al. Co-transplantation of bone marrow-derived endothelial progenitor cells improves revascularization and organization in islet grafts. American journal of transplantation : official journal of the American Society of Transplantation and the American Society of Transplant Surgeons. 2013;13(6):1429-40. Epub 2013/04/23. doi: 10.1111/ajt.12222. PubMed PMID: 23601171.

11. Pepper AR, Gala-Lopez B, Pawlick R, Merani S, Kin T, Shapiro AM. A prevascularized subcutaneous device-less site for islet and cellular transplantation. Nature biotechnology. 2015;33(5):518-23. Epub 2015/04/22. doi: 10.1038/nbt.3211. PubMed PMID: 25893782.

12. Coronel MM, Geusz R, Stabler CL. Mitigating hypoxic stress on pancreatic islets via in situ oxygen generating biomaterial. Biomaterials. 2017;129:139-51. Epub 2017/03/28. doi: 10.1016/j.biomaterials.2017.03.018. PubMed PMID: 28342320; PubMed Central PMCID: PMCPMC5497707.

13. Hamilton DC, Shih HH, Schubert RA, Michie SA, Staats PN, Kaplan DL, et al. A silk-based encapsulation platform for pancreatic islet transplantation improves islet function in vivo. Journal of tissue engineering and regenerative medicine. 2017;11(3):887-95. Epub 2015/01/27. doi: 10.1002/term.1990. PubMed PMID: 25619945.

14. Lee HS, Lee JG, Yeom HJ, Chung YS, Kang B, Hurh S, et al. The Introduction of Human Heme Oxygenase-1 and Soluble Tumor Necrosis Factor-α Receptor Type I With Human IgG1 Fc in Porcine Islets Prolongs Islet Xenograft Survival in Humanized Mice. American journal of transplantation : official journal of the American Society of Transplantation and the American Society of Transplant Surgeons. 2016;16(1):44-57. Epub 2015/10/03. doi: 10.1111/ajt.13467. PubMed PMID: 26430779.

15. Yin N, Han Y, Xu H, Gao Y, Yi T, Yao J, et al. VEGF-conjugated alginate hydrogel prompt angiogenesis and improve pancreatic islet engraftment and function in type 1 diabetes. Materials science & engineering C, Materials for biological applications. 2016;59:958-64. Epub 2015/12/15. doi: 10.1016/j.msec.2015.11.009. PubMed PMID: 26652453.

---

## [Decision Letter · Decision Letter 1]

13 Oct 2020

PONE-D-20-16218R1

A novel pre-vascularized tissue engineered chamber as a site for allogeneic and xenogeneic islet transplantation to establish a bioartificial pancreas

PLOS ONE

Dear Dr. Zhao,

Thank you for submitting your manuscript to PLOS ONE. After careful consideration, we feel that it has merit but does not fully meet PLOS ONE’s publication criteria as it currently stands. Therefore, we invite you to submit a revised version of the manuscript that addresses the points raised during the review process.

We look forward to receiving your revised manuscript.

Kind regards,

Matthias G von Herrath, MD PhD

Academic Editor

PLOS ONE

Reviewers' comments:

Reviewer's Responses to Questions

**Comments to the Author**

1. If the authors have adequately addressed your comments raised in a previous round of review and you feel that this manuscript is now acceptable for publication, you may indicate that here to bypass the “Comments to the Author” section, enter your conflict of interest statement in the “Confidential to Editor” section, and submit your "Accept" recommendation.

Reviewer #1: All comments have been addressed

Reviewer #2: (No Response)

2. Is the manuscript technically sound, and do the data support the conclusions?

Reviewer #1: Yes

Reviewer #2: Partly

3. Has the statistical analysis been performed appropriately and rigorously? 

Reviewer #1: Yes

Reviewer #2: No

4. Have the authors made all data underlying the findings in their manuscript fully available?

Reviewer #1: Yes

Reviewer #2: Yes

5. Is the manuscript presented in an intelligible fashion and written in standard English?

Reviewer #1: Yes

Reviewer #2: No

6. Review Comments to the Author

Reviewer #1: All comments addressed. All comments addressed. All comments addressed. All comments addressed. All comments addressed.

Reviewer #2: Thank you for addressing previous comments.

Regarding statistical analysis, the Authors argue to my previous comment 17 (group size too small) that the use of GraphPad Prism ensures scientific and reasonable data analysis. This is not a valid argument for defending a small group size. A power analysis supporting n=5 in the experimental groups should be shared, since this data is the backbone of the manuscript.

Regarding the language, there are still numerous grammatical errors and unclear sentences throughout the manuscript. As an example, on p 17 top and mid:

“This indicates that maintain euglycemia in recipient mice was dependent on the islets which transplanted into the TEC”

“We could re-implanted xenogeneic islets to the mice which reversal of hyperglycemia had failed, and to explore whether could maintain euglycemia over the long-term following dual anti-CD45RB plus anti-MR-1 antibody treatment.”

I do not feel obliged to provide an exhaustive list but urge the authors to critically and thoroughly revise the language throughout the manuscript.

7. PLOS authors have the option to publish the peer review history of their article (what does this mean?). If published, this will include your full peer review and any attached files.

Reviewer #1: No

Reviewer #2: No

---

## [Author Response · Author response to Decision Letter 1]

17 Nov 2020

Dear editorial-lmanager and Reviewers,

Thank you for providing the valuable suggestions and comments for our manuscript (ID: PONE-D-20-16218). We have carefully revised the manuscript according to the reviewer comments. The answers to the comments are listed as following.

Comment 17. It is concerning, that only 5 animals are included in each group, and I am surprised to see so little variation, since there normally is significant variation among animals. Raw data should be reviewed. Statistical analysis of samples from 5 animals is not sufficiently robust to draw solid conclusions from. Has power analysis been performed? 

Answer: This prevascularized chamber model in murine is really technically demanding and time consuming. And the same model has been also used in islet transplantation in murine groin by different researches [1]. Cronin KJ et.al. used this model to test effects of different materials on revascularization. However, the innovation point in our study is not repeat the previous work which have been proved a successful method to build a prevascularized chamber for islet transplantation, but deepened the work and used this prevascularized chamber model to xenograft transplantation combined with the immunotherapy regime exploited by our research group. Thus, as to a well-documented prevascularized chamber model for islet transplantation, we allocated a small number of valuable model to prove this method could work in allograft transplantation just as that have been well proved by Hussey AJ and Hussey AJ [1], and distributed large number of precious model to test the effect of xenograft transplantation (n=20 in our study). In addition, compared to other works (n=6 in Hussey AJ’s work, and even n=5 in Andrew R Pepper’s work---fig.5 b)[2], the sample size (n=5 in our research) can also prove the efficient of prevascularized chamber in syngeneic and allogeneic transplantation. Thank you again for your careful review and sincere suggestions.

1. Cronin KJ, Messina A, Knight KR, Cooper-White JJ, Stevens GW, Penington AJ, et al. New murine model of spontaneous autologous tissue engineering, combining an arteriovenous pedicle with matrix materials. Plastic and reconstructive surgery. 2004;113(1):260-9. Epub 2004/01/07. doi: 10.1097/01.prs.0000095942.71618.9d. PubMed PMID: 14707645.

2. Pepper AR, Gala-Lopez B, Pawlick R, Merani S, Kin T, Shapiro AM. A prevascularized subcutaneous device-less site for islet and cellular transplantation. Nature biotechnology. 2015;33(5):518-23. Epub 2015/04/22. doi: 10.1038/nbt.3211. PubMed PMID: 25893782.

---

## [Editor Report · Decision Letter 2]

19 Nov 2020

A novel prevascularized tissue-engineered chamber as a site for allogeneic and xenogeneic islet transplantation to establish a bioartificial pancreas

PONE-D-20-16218R2

Dear Dr. Zhao,

We’re pleased to inform you that your manuscript has been judged scientifically suitable for publication and will be formally accepted for publication once it meets all outstanding technical requirements.

Kind regards,

Matthias G von Herrath, MD PhD

Academic Editor

PLOS ONE
---

## [Editor Report · Acceptance letter]

24 Nov 2020

PONE-D-20-16218R2 

A novel prevascularized tissue-engineered chamber as a site for allogeneic and xenogeneic islet transplantation to establish a bioartificial pancreas 

Dear Dr. Zhao:

I'm pleased to inform you that your manuscript has been deemed suitable for publication in PLOS ONE. Congratulations! Your manuscript is now with our production department. 

Kind regards, 

on behalf of

Prof. Matthias G von Herrath 

Academic Editor

PLOS ONE